# Evaluation of Viral Suppression in Paediatric Populations: Implications for the Transition to Dolutegravir-Based Regimens in Cameroon: The CIPHER-ADOLA Study

**DOI:** 10.3390/biomedicines12092083

**Published:** 2024-09-12

**Authors:** Joseph Fokam, Yagai Bouba, Rogers Awoh Ajeh, Dominik Tameza Guebiapsi, Suzane Essamba, Albert Franck Zeh Meka, Ebiama Lifanda, Rose Armelle Ada, Liman Yakouba, Nancy Barbara Mbengono, Audrey Raissa Dzaddi Djomo, Suzie Ndiang Tetang, Samuel Martin Sosso, Jocelyne Carmen Babodo, Olivia Francette Ndomo Ambomo, Edith Michele Temgoua, Caroline Medouane, Sabine Ndejo Atsinkou, Justin Leonel Mvogo, Roger Martin Onana, Jean de Dieu Anoubissi, Alice Ketchaji, Alex Durand Nka, Davy-Hyacinthe Anguechia Gouissi, Aude Christelle Ka’e, Nadine Nguendjoung Fainguem, Rachel Simo Kamgaing, Désiré Takou, Michel Carlos Tommo Tchouaket, Ezechiel Ngoufack Jagni Semengue, Marie Amougou Atsama, Julius Nwobegahay, Comfort Vuchas, Anna Nya Nsimen, Bertrand Eyoum Bille, Sandra kenmegne Gatchuessi, Francis Ndongo Ateba, Daniel Kesseng, Serge Clotaire Billong, Daniele Armenia, Maria Mercedes Santoro, Francesca Ceccherini-Silberstein, Paul Ndombo Koki, Hadja Cherif Hamsatou, Vittorio Colizzi, Alexis Ndjolo, Carlo-Federico Perno, Anne-Cecile Zoung-Kanyi Bissek

**Affiliations:** 1Faculty of Health Sciences, University of Buea, Buea P.O. Box 63, Cameroon; ajehrogers@gmail.com; 2Chantal BIYA International Reference Centre for Research on HIV/AIDS Prevention and Management, Yaoundé P.O. Box 3077, Cameroon; martinsosso@yahoo.it (S.M.S.); nkaalexdurand@yahoo.com (A.D.N.); davygouissi@gmail.com (D.-H.A.G.); kae.audechristelle@gmail.com (A.C.K.); fainguem_dine@yahoo.fr (N.N.F.); r.kamgaing@yahoo.it (R.S.K.); dtakou@yahoo.com (D.T.); tommomichel@yahoo.fr (M.C.T.T.); ezechiel.semengue@gmail.com (E.N.J.S.); andjolo@yahoo.com (A.N.); 3Central Technical Group, National AIDS Control Committee (NACC), Yaoundé P.O. Box 2459, Cameroon; guebiapsitameza@gmail.com (D.T.G.); suzanneessamba@yahoo.fr (S.E.); franckalbert.meka@cnls.cm (A.F.Z.M.); rose.ada@cnls.cm (R.A.A.); yakouba.liman@cnls.cm (L.Y.); barbara.mbengono@cnls.cm (N.B.M.); carmenbabodo@gmail.com (J.C.B.); johannesndomo@yahoo.com (O.F.N.A.); edithtemgoua@yahoo.fr (E.M.T.); caroline.medouane@cnls.cm (C.M.); atndsa2000@gmail.com (S.N.A.); mvogoleonel1991@gmail.com (J.L.M.); rogers.ajeh@cnls.cm (R.M.O.); jeandedieu.anoubissi@cnls.cm (J.d.D.A.); hamsatouhadja@cnls.cm (H.C.H.); 4Faculty of Medicine, UniCamillus—Saint Camillus International University of Health Sciences, 00131 Rome, Italy; daniele.armenia@unicamillus.org; 5HIV, Tuberculosis and Malaria Global Funds Subvention Coordination Unit (UCS), Ministry of Public Health, Yaoundé P.O. Box 2459, Cameroon; 6Health Office, United States Agency for International Development (USAID), Yaoundé P.O. Box 817, Cameroon; elifanda@usaid.gov (E.L.); adjomonzaddi@usaid.gov (A.R.D.D.); 7Essos Hospital (CHE), National Social Welfare Centre, Yaoundé P.O. Box 5777, Cameroon; ndiangsuzie@yahoo.fr; 8Division of Disease, Epidemic and Pandemic Control, Ministry of Public Health, Yaoundé P.O. Box 3038, Cameroon; ketchajialice2015@gmail.com; 9Faculty of Medicine and Biomedical Sciences, University of Yaoundé I, Yaoundé P.O. Box 1364, Cameroon; sergebillong@yahoo.fr (S.C.B.); annezkbissek@yahoo.fr (A.-C.Z.-K.B.); 10Department of Experimental Medicine, University of Rome “Tor Vergata”, 00133 Rome, Italy; santormaria@gmail.com (M.M.S.); ceccherini@med.uniroma2.it (F.C.-S.); 11Research Center on Emerging and Re-Emerging Diseases (CREMER), Yaoundé P.O. Box 13033, Cameroon; marieamougou164@yahoo.com; 12Centre for Research and Military Health (CRESAR), Yaoundé P.O. Box 15939, Cameroon; nwobegahay@yahoo.com; 13The Bamenda Center for Health Promotion and Research, Bamenda P.O. Box 586, Cameroon; vuchascommy@yahoo.com (C.V.); nsimenn@yahoo.com (A.N.N.); 14Retrovirology Laboratory, Laquintinie Hospital, Douala P.O. Box 4035, Cameroon; eyoumbille@yahoo.fr; 15Fondation Sociale Suisse, Pette Hospital, Pette P.O. Box 65, Cameroon; gatch.sk@gmail.com; 16Mother-Child Centre, Chantal BIYA Foundation, Yaoundé P.O. Box 1936, Cameroon; atebfranc@gmail.com (F.N.A.); dkesseng@gmail.com (D.K.); koki_paul@hotmail.com (P.N.K.); 17Faculty of Science and Technology, University of Bandjoun, Bandjoun P.O. Box 127, Cameroon; colizzi@bio.uniroma2.it; 18Multimodal Medicine Laboratory, Bambino Gesù Children Hospital, IRCCS, 00165 Rome, Italy; carlofederico.perno@opbg.net; 19Division of Health Operational Research, Ministry of Public Health, Yaoundé P.O. Box 1937, Cameroon

**Keywords:** viral suppression, low-level viremia, children, adolescents, young adults, DTG-based regimen, elimination of paediatric AIDS, Cameroon

## Abstract

Mortality in children accounts for 15% of all AIDS-related deaths globally, with a higher burden among Cameroonian children (25%), likely driven by poor virological response. We sought to evaluate viral suppression (VS) and its determinants in a nationally representative paediatric and young adult population receiving antiretroviral therapy (ART). A cross-sectional and multicentric study was conducted among Cameroonian children (<10 years), adolescents (10–19 years) and young adults (20–24 years). Data were collected from the databases of nine reference laboratories from December 2023 to March 2024. A conditional backward stepwise regression model was built to assess the predictors of VS, defined as a viral load (VL) <1000 HIV-RNA copies/mL. Overall, 7558 individuals (females: 73.2%) were analysed. Regarding the ART regimen, 17% of children, 80% of adolescents and 83% of young adults transitioned to dolutegravir (DTG)-based regimens. Overall VS was 82.3%, with 67.3% (<10 years), 80.5% (10–19 years) and 86.5% (20–24 years), and *p* < 0.001. VS was 85.1% on a DTG-based regimen versus 80.0% on efavirenz/nevirapine and 65.6% on lopinavir/ritonavir or atazanavir/ritonavir. VS was higher in females versus males (85.8% versus 78.2%, *p* < 0.001). The VS rate remained stable around 85% at 12 and 24 months but dropped to about 80% at 36 months after ART initiation, *p* < 0.009. Independent predictors of non-VS were younger age, longer ART duration (>36 months), backbone drug (non-TDF/3TC) and anchor drug (non-DTG based). In this Cameroonian paediatric population with varying levels of transition to DTG, overall VS remains below the 95% targets. Predictors of non-VS are younger age, non-TDF/3TC- and non-DTG-based regimens. Thus, efforts toward eliminating paediatric AIDS should prioritise the transition to a DTG-based regimen in this new ART era.

## 1. Introduction

HIV is still one of the major public health problems worldwide, especially among children and adolescents, with 3.7 million people living with HIV (PLHIV) in 2020 [1,2]. Of these PLHIV, 1.8 (1.2–2.3) million were adolescents (AVVIH), and around 88% of them were in sub-Saharan Africa (SSA) [3]. The Joint United Nations Programme on HIV/AIDS (UNAIDS) 2019 recognised that the reduction of new HIV infections among children, adolescents and young people is slower when compared to adults. For young people aged 20–24, there are about 6000 new infections every day worldwide, [3] and this age group contains the majority of new HIV infections in SSA. Despite the efforts to limit the spread and effects of the disease, approximately 50% of children worldwide are not on antiretroviral therapy (ART). In Cameroon, the rate of HIV vertical transmission was 15% with suboptimal linkage to ART among people under 25 years of age (44.2%). For adolescents, this rate was 39.7% and 45.1%, respectively, for 10–14 years and 15–19 years, while for young adults aged 20–24 years, a rate of up to 62% was reported [4]. For children under 10 years, linkage to ART was even worse (33%) [4]. In addition, in Cameroon, of the HIV-related deaths recorded in 2020, about 25% occurred among children under 15 years of age [4]. This is due in part to the high levels of virological failure observed in the paediatric population.

In the past years, first-line paediatric regimens were based on first-generation non-nucleoside reverse transcriptase inhibitors (NNRTIs), which have a low genetic barrier to resistance, and the second line on protease inhibitor-based regimen (PI/r). Previous studies in Cameroon showed that the virological failure among children and adolescents was about 24% and 47%, respectively, with declining performance among those on NNRTIs [5]. In fact, optimal response to treatment in the paediatric population and among people living with HIV is still a major issue, which is due to the high risk of poor adherence and resistance to antiretrovirals [5,6,7,8,9,10]. Scientific evidence has shown that in Cameroon around a third of infected children are poorly adherent to treatment, with a difference between urban and rural areas [11]. Several other factors have been demonstrated to be associated with virological response [11,12,13,14]. Also, the suboptimal coverage in viral load results in the late detection of treatment failure with accumulation of drug resistance mutations at rates of up to 80% [11]. This situation might limit the treatment option as children grow up, calling for effective public health strategies for optimal sequencing of treatment regimens to maximise long-term treatment success and reduce AIDS-related deaths.

The armamentarium of HIV drugs has considerably increased in recent years and the introduction of integrase inhibitors (INIs) into this arsenal has played a key role in viral control. Dolutegravir, which is a second-generation integrase inhibitor, approved since 2013, is a potent drug with a high genetic barrier. Today, DTG-based regimens are recommended by the WHO as the preferred regimen for PLHIV starting or switching to an optimal regimen [15]. Several clinical trials have investigated the safety and efficacy of DTG in adults, adolescents and children living with HIV/AIDS. For example, the ODYSSEY (Once daily DTG-based ART in Young people vs. Standard thErapY) and IMPAACT (International Maternal Paediatric Adolescents AIDS Clinical Trials Network) P1093 clinical trials demonstrated that DTG-based regimens are effective in this population [8,16,17,18]. These regimens have been investigated in adults and adolescents for first-line and second-line treatment in several clinical trials [19,20,21,22,23].

Currently, Cameroon is transitioning to a DTG-based regimen for children and adolescents, but challenges exist in the scaling up of a DTG-based regimen, especially among children. In this context, recent studies in Cameroon showed that the virological control is between 74.4% and 88.2% in adolescents [24,25] and 64.8% in children. However, these studies have limited sample size and were geographically limited. Therefore, a study with a representative sample size, covering a much larger area in the country is required. Moreover, data on young adults who also generally represent a fragile population are lacking. Studies have shown that viral suppression and undetectable viremia are low compared to adult ages [26,27]. Mortality is also high among young adults, especially those who were transitioned from paediatric care [28]. These observations might be due to gaps in care and adherence issues among this population [27,29]. Therefore, the objective of this study is to evaluate virological response and to identify factors independently associated with viral suppression among children, adolescents and young adults in the DTG era, using a large, multicentre and nationally representative sample.

## 2. Materials and Methods

### 2.1. Study Design and Inclusion Criteria

Following a nationally representative design, a retrospective cross-sectional study was conducted on viral load (VL) between 2022 and 2023 among children (<10 years), adolescents (10–19 years) and young adults (20–24 years) within the framework of the Cameroon ART programme. This was a multicentre study performed on a large scale with exhaustive sampling of all eligible samples at country level. The inclusion criteria were as follows: (1) age less than 25 years; (2) available information on sex and ART regimens; and (3) available results for HIV-RNA load measurement.

### 2.2. Study Population and Data Collection

The study population was made up of PLHIV <25 years stratified into five groups: (1) children <5 years; (2) children 5–9 years; (3) adolescents 10–14 years; (4) adolescents 15–19 years; and (5) young adults 20–24 years. The data collection was performed from December 2023 to March 2024 in all the ten regions of Cameroon through nine HIV reference laboratories for VL. These laboratories receive samples from all the follow-up units for HIV VL testing in the country. This process was coordinated at the national level by the laboratory unit of the Central Technical Group of the National AIDS Control Committee (NACC). At the regional level, the Regional Technical Groups of NACC supervised the data collection at the reference laboratory of their region. Following the inclusion criteria, the sociodemographic, treatment and viral load data were collected in an anonymised form to ensure patient confidentiality. Given that the data were not initially collected for research purposes, the database was carefully screened and all the patients with missing or conflicting information were eliminated at the level of the data collectors at the sites. The data collected from the original databases at various sites was purposed for this study, as part of the collaborative initiative on paediatric HIV education and research (CIPHER-ADOLA). All the data were then centralized at NACC and double-checked to ensure the quality of data, prior to validation.

### 2.3. Description of Study Variables

Three groups of variables were collected from the various databases. The first set of variables is the sociodemographic data, where sex (male or female) and age were collected. Age was collected in years; stratification was performed to group the patients in the following categories: children (<10 years), adolescents (10–19 years) and young adults (20–24 years). Children and adolescents were further stratified into two groups each, as this may have programmatic implications. The second set of variables is the treatment-related data, where ART duration and the specific regimen were collected. For the treatment duration, information on the timing of viral load, that is, at 6 months, 12 months, 24 months, 36 months or more than 36 months after ART initiation was collected. For each regimen, information on the NRTI backbone as well as the anchor drug were considered. Lastly, we collected viral load (in copies/mL) data for which sociodemographic and treatment data were available. Other data such as whether the patient is living in a rural or an urban setting and the route of HIV transmission were not available for most patients (as the database was primary collected for programmatic purposes), thus, they were not considered in the analyses. Given that the study objective was to evaluate the treatment response, viral load (which was categorised as <1000 copies/mL and ≥1000 copies/mL) was considered as the primary outcome (dependent variable). Sex, age, ART duration, backbone and anchor drug were all considered as independent variables.

### 2.4. Viral Load Measurement

The viral load was measured with a conventional real-time PCR technique. Currently, in Cameroon, VL is performed using either point-of-care technologies or conventional platforms. The samples were collected, centrifuged and aliquoted at various collection sites, and then transported to the reference laboratory for analysis. Testing was performed using a plasma sample. The detection limit is <40 copies/mL for the laboratories using the Abbott m2000 system RealTime HIV-1 Quantitative platform (Abbott Park, IL, USA, https://www.molecular.abbott/us/en/products/instrumentation/m2000-realtime-system, accessed on 6 August 2024) or GeneXpert^®^ HIV-1 Viral (https://www.cepheid.com/fr-CH/tests/blood-virology-womens-health-sexual-health/xpert-hiv-1-viral-load.html, accessed on 6 August 2024), and <390 copies/mL for those using Biocentric open platforms (Bandol, France; https://www.biocentric.com/generic-hiv-charge-virale, accessed on 6 August 2024).

### 2.5. Statistical Analysis

After a data cleaning process in Microsoft Excel 2019, the file was imported into IBM SPSS Statistics for Windows (from version 7 and above), version 26, (SPSS Inc., Armonk, NY, USA) [30]. Data were summarised as proportions using tables and figures. The proportions among the various categories of the variables of interest were compared using the chi-square test, chi-square for trend or Fisher’s exact test as appropriate. A binary logistic regression model was used to identify factors independently associated with VS. The independent variables that were considered in the model were sex, age groups, ART duration, ART treatment line, and nucleoside reverse transcriptase (NRTI) backbone and anchor drugs. A conditional backward stepwise regression model was built to assess the predictors of VS. The statistical significance was set at *p* < 0.05 for all the analyses.

Given the lack of a system to track all the follow-up viral load measurements, in these analyses, we considered only the last available viral load measurement. VS was defined as an HIV-RNA measurement of <1000 copies/mL (according to the WHO guidelines), after at least six months from the start of treatment. Because several platforms with varying detection thresholds (including point-of-care technologies) were used, we have arbitrarily defined low-level viremia in this study as a VL between 400 and 999 copies/mL; and high-level viremia as a VL measurement of ≥100,000 copies/mL.

### 2.6. Ethical Considerations

This study was conducted in accordance with the principles of the Declaration of Helsinki and national regulations. This study received all the required administrative authorisations as per the national regulations. This study presents the results for the baseline assessment of the Collaborative Initiative for Paediatric HIV Education and Research (CIPHER)-ADOLA study, which aims to accelerate an evidence-informed, human rights-based and integrated HIV response for infants, children, adolescents and young people living with and affected by HIV. This study received an ethical clearance (CE N° 0056 CRERSHC/2023) from the ethics committee of the regional delegation of public health for the Centre region. Anonymity was ensured through the entire study process and investigators had no access to information that could identify individual participants during or after data collection.

## 3. Results

### 3.1. Population and Treatment Characteristics

Overall, 7558 individuals (children [11.8%]), adolescents [31.7%] and young adults [56.5%]) were analysed. Of note, up to 326 (about 5%) of the patients had less than 5 years of age. Most of the patients were female (73.2%). The proportion of females significantly increased with increasing age, reaching up to 82.2% in the age group 20–24 years (Table 1).

Regarding the ART regimen, all the participants were treatment experienced, with about 44% being on ART for more than 3 years. According to ART regimen line, 89.9%, 10.0% and 0.1% were on first line, second line and third line, respectively. About 67% of patients on second line had <10 years of age (Table 1). Regarding the specific regimens at the time of testing, the most frequent was tenofovir/lamivudine/dolutegravir [TDF/3TC/DTG] (70.8%), followed by TDF/3TC/efavirenz [EFV] or -nevirapine [NVP] (12.9%). Some patients were on second-line treatment with a protease inhibitor-based regimen; notably abacavir/lamivudine/atazanavir/ritonavir or lopinavir/ritonavir [ABC/3TC/ATV/r or LPV/r] (7.0%). A proportion of about 88% of children <10 years were on ABC/3TC/ATV/r or LPV/r.

Regarding the NRTI backbone, as expected, most of the patients received TDF/3TC (86.1%), followed by ABC/3TC (12.7%). Finally, concerning the anchor drug, about 17% of children and about 80% of adolescents received DTG, while for young adults 20–24 years, 82.9% received DTG. Those receiving a DTG-based regimen significantly increased with increasing age, while LPV/r- or ATV/r-based regimens significantly decreased with increasing age (Table 1).

### 3.2. Viral Suppression among Children, Adolescents and Young Adults According to Sex and Age

The overall VS (defined as a viral load measurement <1000 copies/mL) was 82.3% (95% CI: 81.5–83.2). According to sex, a significantly higher VS was observed in females (83.8% [82.8–84.9]), when compared to males (78.2% [76.4–80.0]), *p* < 0.001.

According to ages groups, we found that the VS (95% CI) among children (<10 years), adolescents (10–19 years) and young adults (20–24 years) was 67.4% (64.3–70.5), 80.7% (79.0–82.2) and 86.2% (85.2–87.2), respectively, *p* < 0.001 (Figure 1). When further stratifying the age group <10 years into <5 years and 5–9 years, we found that those <5 years showed a significantly lower VS (61.0% [95% CI: 55.7–66.2]) when compared to 5–9 years (70.9% [95% CI: 67.1–74.6]), *p* = 0.002. However, when stratifying the adolescent group, a similar VS was observed between those aged 10–14 (79.2% [76.6–81.6]) and those aged 15–19 years (81.4% [79.3–83.3]), *p* = 0.192. Figure 1 shows that the rate of viral non-suppression (VnS) significantly decreases with increasing age of the patients.

### 3.3. Viral Suppression According to Treatment Parameters

Our analysis showed that the proportion of those at about 6 months after treatment initiation was similar between VS and VnS groups (Table 2). However, the proportion of those at 24 months was significantly higher among the VS group (13.9%) versus the VnS group (11.2%), *p* = 0.008. Moreover, the proportion of patients receiving treatment for more than 36 months was higher in the VnS group (47.3%), when compared to the VS group (43.7%), *p* = 0.017. According to ART line, the proportion of patients on first line was significantly higher in the VS group (91.9%) versus the VnS group (80.4%), *p* < 0.001.

The proportion of those receiving TDF/3TC/DTG was significantly higher among cases in VS (73.5%), compared to those experiencing VnS (58.1%), *p* < 0.001, while those receiving TDF/3TC/EFV or NVP had similar proportions in VS and VnS (13.0% versus 12.6% versus 13.0%, respectively), *p* = 0.670. Regarding the third 95% UNAIDS target, the regimen with the highest VS level was TDF/3TC/DTG (4575/535, 85.5% [84.5–86.4]), followed by TDF/3TC/EFV (810/978, 82.8% [80.3–85.1]). A PI-based regimen such as ABC/3TC/ATV/r or LPV/r had a higher proportion in VnS (14.2%) versus 5.4% in VS, *p* < 0.001.

When considering the NRTI backbone, TDF/3TC had the highest VS rate (84.8% [83.9–85.6]), compared to ABC/3TC (67.2% [64.2–70.1]) and AZT/3TC (67.4 [57.2–76.5]), *p* < 0.001. Finally, according to anchor drug, DTG showed the highest VS rate (85.1 [84.1–86]), followed by EFV or NVP (80.0% [77.6–82.1]). ATV or LPV showed the highest non-suppression rate (34.4%).

### 3.4. Levels of Viral Load among Children, Adolescents and Young Adults

According to viral load levels and age, Figure 2 shows an increasing proportion of those with VL <400 copies/mL (including those that are undetectable) with increasing age: the lowest being observed in those <5 years (56.1%) and the highest in those aged 20–24 years (82.4%), *p* < 0.001. When looking at the proportion of those with low-level viremia (LLV, defined here as HIV-RNA between 400 and 999 copies/mL), children, adolescents and young adults showed a proportion of 5.9%, 3.5% and 4.1%, respectively, *p* < 0.001.

Looking at LLV according to anchor drug, we found that it was highest among those receiving a PI-based regimen (5.5%), followed by a NNRTI-based regimen (4.8%), with a DTG-based regimen being the lowest (3.7%), *p* = 0.030. Interestingly, the overall proportion of those failing (HIV-RNA >1000 copies/mL) with very high-level viremia (HLV, >100 000 HIV-RNA copies/mL) was 21.1%. Of these, 20.4%, 16.9% and 27.5% was observed among those receiving DTG, EFV or NVP, and ATV or LPV, *p* = 0.010. Furthermore, our analysis evidenced that the proportion of patients with very HLV at failure was significantly higher among those <5 years (11.7%), and this proportion significantly decreased with increasing age (Figure 2).

### 3.5. Factors Associated with Viral Suppression among Children, Adolescents and Young Adults

The regression model (Table 3) was built using all the variables in Table 2. At the univariate level, we found that sex, age, ART duration, ART line and ART regimens were confounders. After adjusting for all these variables, only age, ART duration and ART regimen were found to be independently associated with VS. In particular, for age, when compared to those 20–24 years, age groups <5 years (aOR [95% CI]: 0.469 [0.331–0.665]), 5–9 years (aOR [95% CI]: 0.702 [0.511–0.965]), 10–14 years (aOR [95% CI]: 0.685 [0.564–0.833]) and 15–19 years (aOR [95% CI]: 0.722 [0.612–0.851]) had significantly lower odds of VS. Regarding ART duration, compared to those on ART for >36 months, those on ART for 24 months showed a significantly higher odds of VS (aOR [95% CI]: 1.271 [1.037–1.557]). Regarding NRTI backbone with TDF/3TC, those receiving ABC/3TC and AZT/3TC showed a significantly lower odds of VS. Regarding anchor drug, those receiving an EFV/NVP- or ATV/LPV/DRV/r-containing regimen had about 1.3 and 10.5 lower odds of VS.

Following regression analysis considering each sub-population, for children, the only predictor of VS was age, with children <5 years showing a significantly lower odds of VS (aOR [95% CI]: 0.596 [0.433–0.819], *p* = 0.001), when compared to children 5–9 years (Appendix A). Among adolescents, NRTI backbone and anchor drug were independent predictors of VS. In particular, those receiving DTG had 1.3 and 2.0 higher odds of VS, compared to EFV and PI/r, respectively, (Appendix A). For young adults, those receiving a DTG-based regimen had 2.3 higher odds of VS, when compared to a PI/r-based regimen; while no significant difference was found when compared to EFV-based regimens.

## 4. Discussion

According to the new global alliance, a key pillar for ending AIDS in children by 2030 is the access for children and adolescents to optimized treatment. Limited evidence from SSA calls for a large-scale investigation of viral suppression in children, adolescents and young adults in Cameroon considering sociodemographic and therapeutic features. To the best of our knowledge, this is the first study that reports VS in a nationally representative sample of children and adolescents, collected over Cameroon’s national territory. It should be noted that Cameroon is currently in the transition phase to DTG-based regimens also for children and adolescents. The result of our study shows that viral suppression among children, adolescents and young adults in Cameroon is suboptimal (about 82%), which is below the global UNAIDS target of 95% by 2025, with a performance of 67.3%, 80.5% and 86.5% among children, adolescents and young adults, respectively. VS was highest among patients receiving a DTG-based regimen (85.1%), followed by EFV/NVP 80.0%. Globally, independent predictors of VnS were younger age, longer ART-duration (>36 months), NRTI backbone (non-TDF/3TC) and anchor drug (non-DTG).

A recent study in Cameroon by our team (Fokam et al. [24]) showed that the overall (including adults) VS rate was 89.8%, which is higher compared to the rate observed in our present study. Furthermore, the VS rate observed among the subgroup of young adults in our study is similar to the data reported by Fokam et al. This may be partly explained by the fact that, even though some young adults still struggle with problems such as ART adherence and stigma, most of them have already transitioned to an optimal ART regimen, notably, TDF/3TC/DTG (TLD). This is further supported by the proportion of young adults on TLD in our study that was more than 80% (Table 1).

On the other hand, the VS rate was much lower among children and adolescents, where the VS rates were, respectively, about 67% and 81%. In fact, we have seen that children <5 years recorded the worst VS rate, at only 61%. The study by Fokam et al. reported a lower but similar VS among children [24]. The management of ART treatment in the paediatric populations is complex, with sometimes complex formulations. This might partly justify the poor virological control among children. Compared to adult populations, children are usually receiving a suboptimal regimen or have limited paediatric treatment options. In this study, the proportion of children <5 years and 5–9 years on a PI/r-based regimen (second line) was 68% and 44%, respectively. This poor treatment outcome on PI/r-based regimens is mainly driven by previous exposure to a first-line regimen, which indicates suboptimal regimens on the second line due to the use of NRTI affected by HIV drug resistance. In many low-and middle-income countries (LMICs), there is limited access to optimal paediatric formulations and insufficient training of healthcare workers in the monitoring of children living with HIV. In this regard, the creation of paediatric centres of excellence for HIV in Cameroon and other LMICs will hopefully contribute to reducing this gap. The transition from the current to a more potent regimen based on paediatric DTG is slowly ongoing but needs a rapid scale-up in Cameroon to optimise VS in paediatrics. The findings showing that a DTG-based regimen was a positive predictor of VS is a call to rapidly transition to DTG-based regimens in Cameroon and similar settings, irrespective of the child age groups. This is mainly attributed to the higher genetic barrier and higher potency of the viral replicative capacity [24].

Concerning adolescents, compared to the data of previous years which showed a VS of about 53% [5], the VS has significantly increased and this study indicate a performance of about 81%. Despite this progress, the improvement of viral control among children and adolescents is much slower compared to adults. A large-scale study among children and adolescents showed that the viral suppression was 36% at 1 year, 30% at 2 years and 24% at 3 years after ART initiation [31]. This low performance could be explained by the fact that data were collected in the non-DTG era. The VS among adolescents in this study is also slightly different from the data reported in recent studies by Fokam et al. (74%) [24] and Djiyou et al. (89%) [25]. This difference might be attributed to the difference in sample size and national representativeness of the study sample size. The improvement in VS among adolescents can be attributed to many factors, among which the transition to DTG-based regimens, the psychological support and the creation of social support groups have had a huge impact on ART adherence. In their study assessing the impact of adherence on viral suppression, Tanyi et al. showed that viral suppression was better when adherence to treatment was good [23]. A study conducted in Cameroon also demonstrated that virologic response appears to be better in adolescents who followed therapeutic education sessions [32]. Therefore, achieving the 95% VS target by 2025 in this group will require continuous efforts in the scale-up of DTG-based regimens [24], while providing appropriate social and psychological support strategies. The approval of long-acting cabotegravir plus rilpivirine for eligible adolescents represents an additional opportunity to improve VS. It is, therefore, important to provide some evidence (mostly virological, i.e., viral load and resistance) on the possible introduction of long-acting drugs in this fragile population, which struggles with some issues related to adolescence. In this regard, the CIPHER-ADOLA study, which aims at optimising the follow-up of HIV-infected adolescents, will fill some gaps. Enough literature exists on the potency of DTG-based regimens [17], even among those who were previously exposed to boosted protease inhibitors. Even though a drug resistance test was not performed in this study, previous reports indicated that there is a low ART switch rate from one line to the other. Previous studies reported an alarming levels of drug resistance among the adolescent populations in Cameroon [11]. This means that reliable access to viral load testing is important and, therefore, should be improved to confirm virological failure and proceed with the switch to other therapeutic lines. Efforts should also be made to improve the access to resistance testing, especially in paediatric populations.

Regarding the factors associated with VS among children, adolescents and young adults in Cameroon, we have found that age, NRTI backbone, anchor drug and ART duration were independently associated with VS. This data has been observed in studies conducted in a similar setting [24,25,33]. Compared to patients that received a DTG-based regimen, those that received an EFV-based regimen had only 1.3 higher odds of VS. Despite this fact, if the patient is eligible for DTG, the WHO recommends switching to DTG because of the higher durability of VS with a DTG-based regimen and the preservation of the NRTI backbone from resistance [15]. Moreover, a review of relevant studies comparing DTG to EFV found that, DTG had improved odds of VS across all time points (OR: 1·94 [1·48–2·56] at 96 weeks) [34]. In Cameroon, the NAMSAL clinical trial found that, in HIV-1-infected adults in Cameroon, a DTG-based regimen was not inferior to an EFV400-based reference regimen with regard to VS at week 48 (DTG versus EFV: 74.5% versus 69.0) [35]. In general, several studies have demonstrated that DTG is associated with a higher virological response rate than boosted protease inhibitors, including darunavir [20,21]. Similar to the data observed in adults patients, DTG plus an optimized background regimen seemed safe, well tolerated and efficacious among treatment-experienced HIV-1-infected adolescents [8]. The data from the present study are in line with these reports, therefore favoring the scale up to a DTG-based regimen in the paediatric population. This result should be considered and confirmed in future large-scale clinical trials and cohort studies. In the paediatric population, alongside the transition to a DTG-based regimen, one important intervention that remains critical in the treatment success is the provision of ART adherence support. Concerning sex, unlike in other studies, it was not found to be a predictor of VS in our study population.

Beyond VS, our results evidenced that in our setting, a high proportion (>20%) of virally non-suppressed patients failed their ART with very high-level viremia. Additionally, in this population, we have seen that the proportion of those who are virologically suppressed (<1000 HIV-RNA copies/mL) but showed a low-level viremia is not negligible. LLV is now a rising problem, and appropriate measures should be taken to reduce the extent of this phenomenon. Of note, those who failed their ART with LLV were less common in those receiving a DTG-based regimen. However, for high-level viremia, we did not see any difference in terms of anchor drug. More studies are warranted to better understand the HIV viral dynamics in the paediatric population.

This study has some limitations that should be overcome in subsequent analysis. First, the database was not primarily designed for research purposes, therefore, some variables, such as education level and living in rural settings as opposed to urban settings, were not collected from the patients. Also, being collected for routine and programmatic purposes, not primarily for research purposes, there is a possibility that selection biases were introduced. Secondly, resistance test results and adherence data that could have helped to better characterise the reasons for VnS were not available. However, appropriate statistical methods were implemented to make corrections, whenever necessary. The major strength of this study is that analyses were performed on a large sample size collected from all the ten regions of Cameroon, thus being nationally representative.

## 5. Conclusions

In this Cameroonian paediatric population with varying levels of transition to DTG (20% in children and 80% in adolescents), overall VS (82.3%) remains below the 95% UNAIDS targets, especially among children <10 years (67.4%). Interestingly, predictors of non-VS are younger age and non-DTG-based regimens. Thus, efforts toward eliminating paediatric AIDS should prioritise transition to a DTG-based regimen, especially for adolescents in this new ART era.

## Figures and Tables

**Figure 1 biomedicines-12-02083-f001:**
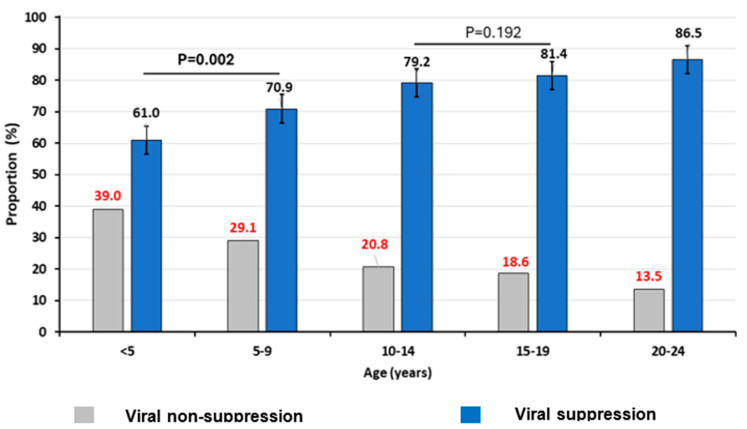
Viral suppression according to age categories. Comparisons were made using chi-square tests. Viral suppression was defined as an HIV-RNA measurement <1000 copies/mL.

**Figure 2 biomedicines-12-02083-f002:**
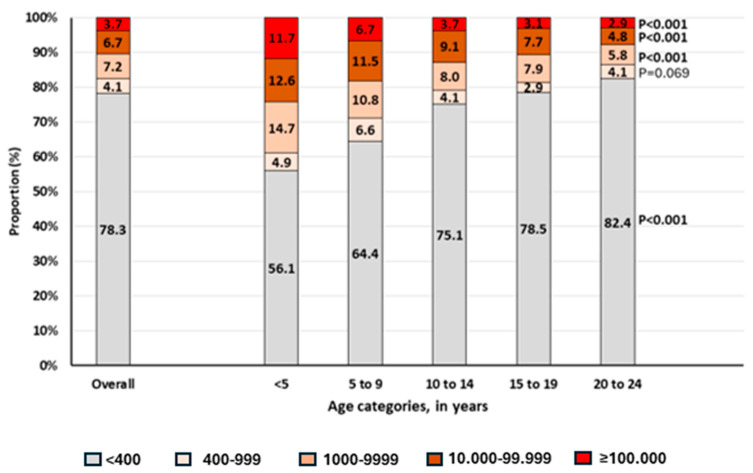
Viral load measurement levels stratified according to age. Viral load measurement is presented in copies/mL. A viral load level <400 copies/mL includes those that are undetectable. *p*-Values were computed using chi-square for trend tests.

**Table 1 biomedicines-12-02083-t001:** Sociodemographic and treatment characteristics according to age categories.

Characteristics	Total (n = 7558)	Age Categories, Years				
		<5 (n = 326, 4.4%)	5–9 (n = 564, 7.5%)	10–14 (n = 1001, 13.2%)	15–19 (n = 1395, 18.5%)	20–24 (n = 4272, 56.5%)
Sex, n (%)						
*Male*	2027 (26.8)	171 (52.5)	265 (47.0)	452 (45.2)	405 (29.0)	734 (17.2)
*Female*	5531 (73.2)	155 (47.5)	299 (53.0)	549 (54.8)	990 (71.0)	3538 (82.8)
Treatment duration in month, n (%)						
*6*	1520 (20.1)	86 (26.4)	56 (9.9)	74 (7.4)	262 (18.8)	1042 (24.4)
*12*	1335 (17.7)	75 (23.0)	75 (13.3)	64 (6.4)	190 (13.6)	931 (21.8)
*24*	1012 (13.4)	71 (21.8)	64 (11.3)	96 (9.6)	145 (10.4)	636 (14.9)
*36*	338 (4.5)	22 (6.7)	34 (6.0)	33 (3.3)	52 (3.7)	197 (4.6)
*>36*	3353 (44.4)	72 (22.1)	335 (59.4)	734 (73.3)	746 (53.5)	1466 (34.3)
ART regimen line, n (%)						
*First line*	6792 (89.9)	93 (1.4)	292 (4.3)	913 (13.4)	1304 (19.2)	4190 (61.7)
*Second line*	755 (10.0)	233 (30.9)	271 (35.9)	87 (11.5)	88 (11.7)	76 (10.1)
*Third line*	11 (0.1)	0 (0)	1 (9.1)	1 (9.1)	3 (27.3)	6 (54.5)
Regimen, n (%)						
*TDF/3TC/DTG*	5351 (70.8)	0 (0.0)	0 (0.0)	705 (13.2)	1113 (20.8)	3533 (66.0)
*ABC/3TC/ATV/r or LPV/r*	528 (7.0)	221 (41.9)	246 (46.6)	36 (6.8)	15 (2.8)	10 (1.9)
*ABC/3TC + DTG*	230 (3.0)	12 (5.2)	137 (59.6)	75 (32.6)	4 (1.7)	2 (0.9)
*TDF/3TC/EFV or NVP*	978 (12.9)	42 (4.3)	37 (3.8)	84 (8.6)	177 (18.1)	638 (65.2)
*Others*	471 (6.3)	51 (10.8)	144 (30.6)	101 (21.4)	86 (18.3)	89 (18.9)
Backbone, n (%)						
*ABC/3TC*	964 (12.7)	272 (28.2)	493 (51.2)	145 (15.1)	27 (2.8)	26 (2.7)
*AZT/3TC*	89 (1.2)	8 (8.9)	26 (28.9)	34 (37.8)	15 (16.7)	7 (7.8)
*TDF/3TC*	6505 (86.1)	46 (0.7)	45 (0.7)	822 (12.6)	1353 (20.8)	4239 (65.2)
Anchor drug, n (%)						
*DTG*	5600 (74.1)	12 (0.2)	141 (2.5)	785 (14)	1121 (20)	3541 (63.2)
*NNRTIs (EVF, NVP)*	1207 (16.0)	81 (6.7)	155 (12.8)	132 (10.9)	186 (15.4)	653 (54.1)
*PIs (AT* *V/r, LPV/r* *, DRV)*	751 (9.9)	233 (31)	268 (35.7)	84 (11.2)	88 (11.7)	78 (10.4)

3TC: lamivudine; ABC: abacavir; ART: antiretroviral treatment; ATV: atazanavir; AZT: zidovudine; DTG: dolutegravir; EFV: efavirenz; LPV: lopinavir; NVP: nevirapine; TDF: tenofovir disoproxil fumarate.

**Table 2 biomedicines-12-02083-t002:** Viral suppression according to treatment parameters.

Characteristics	Overall N = 7558	Viral Suppression Rate		
		Non-Suppressed N = 1335 (17.7)	Suppressed N = 6223 (82.3%)	*p*-Value
Treatment duration, n (%)				
*6*	1520 (20.1)	278 (20.8)	1242 (20.0)	**0.474**
*12*	1335 (17.7)	211 (15.8)	1124 (18.0)	0.050
*24*	1012 (13.4)	149 (11.2)	863 (13.9)	0.008
*36*	338 (4.5)	66 (4.9)	272 (4.4)	0.358
*>36*	3353 (44.3)	631 (47.3)	2722 (43.7)	0.019
ART regimen line, n (%)				
*First line*	6792 (89.9)	1074 (80.4)	5718 (91.9)	<0.001
*Second line*	755 (10.0)	255 (19.1)	500 (8.0)	<0.001
*Third line*	11 (0.1)	6 (0.5)	5 (0.1)	0.006
Regimen, n (%)				
*TDF/3TC/DTG*	5351 (70.8)	776 (58.1)	4575 (73.5)	**<0.001**
*ABC/3TC-ATV/r or LPV/r*	528 (7.0)	190 (14.2)	338 (5.4)	<0.001
*ABC/3TC + DTG*	230 (3.0)	57 (4.3)	173 (2.8)	0.004
*TDF/3TC-EFV/NVP*	978 (13.0)	168 (12.6)	810 (13.0)	0.670
*Others*	471 (6.2)	144 (10.8)	327 (5.3)	<0.001
Backbone, n (%)				
*TDF/3TC*	6505 (86.1)	990 (74.1)	5515 (88.6)	**<0.001**
*ABC/3TC*	964 (12.7)	316 (23.7)	648 (10.4)	**<0.001**
*AZT/3TC*	89 (1.2)	29 (2.2)	60 (1.0)	**<0.001**
Anchor, n (%)				
*DTG*	5600 (74.1)	835 (62.6)	4765 (76.6)	**<0.001**
*EFV or NVP*	1207 (16.0)	242 (18.1)	965 (15.5)	0.018
*ATV/r or LPV/r*	751 (9.9)	258 (19.3)	493 (7.9)	**<0.001**

3TC: lamivudine; ABC: abacavir; ART: antiretroviral treatment; ATV: atazanavir; AZT: zidovudine; DTG: dolutegravir; EFV: efavirenz; LPV: lopinavir; NVP: nevirapine; TDF: tenofovir disoproxil fumarate, r: ritonavir. *p*-Values were computed using chi-square tests.

**Table 3 biomedicines-12-02083-t003:** Factors associated with viral suppression among children, adolescent, and young adults.

Factors	Regression Model			
	Crude OR (95% CI)	*p*-Value	Adjusted OR (95% CI)	*p*-Value
Sex				
*Males*	0.693 (0.610–0.788)	**<0.001**	0.903 (0.787–1.036)	0.145
*Females*	1		1	
Age categories, years				
*<5*	0.244 (0.192–0.310)	**<0.001**	0.521 (0.367–0.741)	**<0.001**
*5–9*	0.380 (0.311–0.465)	**<0.001**	0.764 (0.552–1.057)	**0.104**
*10–14*	0.594 (0.498–0.709)	**<0.001**	0.694 (0.571–0.844)	**<0.001**
*15–19*	0.680 (0.579–0.799)	**<0.001**	0.717 (0.608–0.845)	**<0.001**
*20–24*	1		1	
Treatment duration, months				
*6*	1.036 (0.886–1.211)	0.660	0.914 (0.774–1.079)	0.287
*12*	1.235 (1.041–1.465)	**0.015**	1.093 (0.912–1.309)	0.335
*24*	1.343 (1.106–1.631)	**0.003**	1.280 (1.046–1.567)	**0.017**
*36*	0.955 (0.720–1.267)	0.751	0.926 (0.692–1.239)	0.606
*>36*	1		1	
ART regimen line, n (%)				
*First line*	1		1	
*Second line*	0.368 (0.312–0.434)	**<0.001**	6.731 (0.654–69.311)	0.109
*Third line*	0.157 (0.048–0.514)	**<0.001**	0.676 (0.071–6.450)	0.733
Backbone				
*TDF/3TC*	1		1	
*ABC/3TC*	0.368 (0.316–0.427)	**<0.001**	0.618 (0.464–0.822)	**0.001**
*AZT/3TC*	0.378 (0.241–0.591)	**<0.001**	0.590 (0.358–0.971)	**0.038**
Anchor drug				
*DTG*	1		1	
*EFV/NVP*	0.698 (0.595–0.818)	**<0.001**	0.808 (0.684–0.956)	**0.013**
*ATV/LPV/DRV/r*	0.333 (0.282–0.394)	**<0.001**	0.095 (0.009–0.972)	**0.047**

3TC: lamivudine; ABC: abacavir; ART: antiretroviral treatment; ATV: atazanavir; AZT: zidovudine; DTG: dolutegravir; EFV: efavirenz; LPV: lopinavir; NVP: nevirapine; TDF: tenofovir disoproxil fumarate; OR: odds ratio; TDF: tenofovir disoproxil fumarate. For the multivariate analysis, the model was adjusted for sex, age, treatment duration, ART line, NRTI backbone and anchor drug. *p*-Values in boldface indicates those that were significantly associated (*p* < 0.05) with viral suppression. CI: confidence interval. cART: combined antiretroviral therapy.

## Data Availability

Data are available upon request from the corresponding authors.

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
