# Peer review of "Evaluation of Viral Suppression in Paediatric Populations: Implications for the Transition to Dolutegravir-Based Regimens in Cameroon: The CIPHER-ADOLA Study"

_biomedicines, 2024, doi:10.3390/biomedicines12092083_

Round 1
Reviewer 1 Report
Comments and Suggestions for Authors
ART for children is an important issue especially in developing countries. In this article, the authors analyzed the ART and viral suppression in Cameroon children, adolescents and young adults, particularly emphasized the DTG-containing regimen. This cross-sectional study included 7558 participants who had received ART for at least 6 months. The authors concluded that DTG-based regimen should be prioritized for young HIV-infected children. However, I have some concerns on the data analysis.
1. The conclusion mainly obtained from the results showed in Table 3 by multivariable analysis. However, for ART regimen, the authors took TDF/3TC/DTG as reference, and the adjusted OR of other regimens were based on comparing to TDF/3TC/DTG. The aOR of ABC/3TC/DTG was 0.684 (P 0.046), and TDF/3TC-EFV/NVP was 0.851 (P 0.090). For ABC/3TC/DTG, the difference between the two regimens was TDF and ABC but not DTG, so I will think the aOR of 0.684 may show a less efficiency of ABC than TDF but not DTG. On the other hand, the difference between TDF/3TC-EFV/NVP and TDF/3TC/DTG was really EFV/NVP vs. DTG, but the P value was 0.09 (> 0.05), not statistically significant, which showed no significant VS difference between these two regimens (though there may be a tendency of less effectiveness of EFV/NVP than DTG). The results in Table 2 shows that anchor EFV or NVP was not good as DTG for the viral suppression (80% vs. 85.1%). Based on your analysis, it seems that TDF is more important than DTG? Thus, the authors may need to reanalyze the data (especially consider the category of the regimen or drugs), and then make the conclusion carefully.
2. This study tried to obtain information on the DTG-containing regimen for children in Cameroon. However, in Table 1, there were not children (<10 years) received TDF/3TC/DTG regimen. So I wonder how this situation will influence the overall multivariable analysis in Table 3 and the conclusion.
3. After multivariable analysis for all participants, I recommend the authors to do the stratified multivariable analysis (for each age group).
4. In Table 1, Regimen others/unknown, but there was no unknown in the Backbone and Anchor drug, do you really have unknown regimen? Besides, I recommend you make the proportion in each group (vertical) but not some in vertical some in horizontal. DRV needed to be detailed in footnote.
5. Table 1 is very good for understanding the situation of each group of the participants. I recommend that authors to make Table 2 as Table 1, showing the viral suppression for each group of participants.
6. For easy understanding and following the story, I recommend the authors to give a short summary of your results (discovery) in the first paragraph of discussion and then start discussion.
Minor issues:
1. Line 285, more than 3200 patients. What does this number mean here?
2. Lines 291-292, The prevalence observed among young adults in our population is close to the VS rate observed in this study. This sentence is not clear.
3. The font of the footnote of Table 3 should be same.
4. Fig 1. ≧ should be <.
5. Line 268, ATT should be ATV.
6. Line 302, 5-19 should be 5-9.
7. Line 312, comma should be after [31].
8. Line 318, period should be after [26].
Comments on the Quality of English LanguageEnglish can be read but there were some mistakes. So minor editing is needed.
Author Response
REVIEWER 1
Major
1. The conclusion mainly obtained from the results showed in Table 3 by multivariable analysis. However, for ART regimen, the authors took TDF/3TC/DTG as reference, and the adjusted OR of other regimens were based on comparing to TDF/3TC/DTG. The aOR of ABC/3TC/DTG was 0.684 (P 0.046), and TDF/3TC-EFV/NVP was 0.851 (P 0.090). For ABC/3TC/DTG, the difference between the two regimens was TDF and ABC but not DTG, so I will think the aOR of 0.684 may show a less efficiency of ABC than TDF but not DTG. On the other hand, the difference between TDF/3TC-EFV/NVP and TDF/3TC/DTG was really EFV/NVP vs. DTG, but the P value was 0.09 (> 0.05), not statistically significant, which showed no significant VS difference between these two regimens (though there may be a tendency of less effectiveness of EFV/NVP than DTG). The results in Table 2 shows that anchor EFV or NVP was not good as DTG for the viral suppression (80% vs. 85.1%). Based on your analysis, it seems that TDF is more important than DTG? Thus, the authors may need to reanalyze the data (especially consider the category of the regimen or drugs), and then make the conclusion carefully.
Response:
We thank the reviewer for the time taken to review our manuscript.
As suggested by the reviewer, we have re-analysed the data. To respond also to the concern regarding TDF (part of the backbone) being more important than DTG (anchor drug), we have used in regression models the variables backbone (2 of the nucleoside reverse transcriptase inhibitors, TDF, 3TC, AZT in the Cameroonian context) and anchor drug (commonly EFV, ATV, LPV or DTG), instead of the regimen as it was before.
From Table 3, it can be clearly seen that, TDF/3TC has a significantly higher odds compared to other backbones; and DTG also has a significantly higher odds compared to other anchor drugs.
Therefore, because TDF-backbone was used since several years in the Cameroonian context, and that the newer drug is DTG (for which limited data exist because of recent introduction), we emphasised the switch to DTG in our conclusions. However, we have softened the conclusions in this revised version.
2.This study tried to obtain information on the DTG-containing regimen for children in Cameroon. However, in Table 1, there were not children (<10 years) received TDF/3TC/DTG regimen. So, I wonder how this situation will influence the overall multivariable analysis in Table 3 and the conclusion.
Response:
As already discuss in point one, TDF/3TC/DTG is not recommended for children, therefore it was expected that children <10 years will not receive this regimen. Concerning this aspect and other worries raised in point 1, in this new version, we have performed the analysis considering the backbone and the anchor drug (instead of the complete regimen) to avoid this problem.
3. After multivariable analysis for all participants, I recommend the authors to do the stratified multivariable analysis (for each age group).
Response:
As recommended by the reviewer we have performed the multivariate analysis for each group, and we suggest adding this table as Supplementary tables (see Suppl. Table 1, 2 &3).
This sub analysis shows that among children the only predictor of VS was age; meanwhile among adolescents and young adults, treatment was predictor of VS.
We have added some sentences regarding these results in the part of regression model (see line 334-341).
4. In Table 1, Regimen others/unknown, but there was no unknown in the Backbone and Anchor drug, do you really have unknown regimen? Besides, I recommend you make the proportion in each group (vertical) but not some in vertical some in horizontal. DRV needed to be detailed in footnote
Response:
We thank the reviewers for this comment. All the patients with no treatment have been removed, therefore, it is only “other”. This has been corrected in the Tables.
5. Table 1 is very good for understanding the situation of each group of the participants. I recommend that authors to make Table 2 as Table 1, showing the viral suppression for each group of participants.
Response:
Table 2 is now presented as Table 1, using column proportions. We have also made some changes in the paragraph describing the results of Table 2, to reflect the data in the Table 2.
6. For easy understanding and following the story, I recommend the authors to give a short summary of your results (discovery) in the first paragraph of discussion and then start discussion.
Response:
Response:
We thank the reviewer for this comment. In this new version, we have added some sentences that summarize the main results of this study in the 1st paragraph of the discussion section (see 350-356).
Minor comments:
1. Line 285, more than 3200 patients. What does this number mean here?
Response:
This was to emphasize the important sample size of children and adolescents in the study, but we have deleted the piece “more than 3200 patients” as it might create some confusion for the reader.
2. Lines 291-292, The prevalence observed among young adults in our population is close to the VS rate observed in this study. This sentence is not clear.
Response:
We have rephrased the sentence as follows: “The viral suppression rate observed among the subgroup of young adults in our study is similar to the data reported by Fokam et al.”
3. The font of the footnote of Table 3 should be same.
Response:
The foot note of table 3 has now been adjusted.
4. Fig 1. ≧should be <.
Response:
Thank you for this observation. This has been corrected.
5. Line 268, ATT should be ATV.
Response:
ATT has been changed in ATV
6. Line 302, 5-19 should be 5-9
Response:
Thank you for this observation. “5-19” has been changed to “5-9”.
7. Line 312, comma should be after [31].
Response:
This has been corrected,
8. Line 318, period should be after [26].
Response:
This has been corrected.
Reviewer 2 Report
Comments and Suggestions for Authors
Overall, this research topic is valuable for Biomedicines readers. It created valuable research evidence with the potential to inform the interventions focusing HIV prevention and control. The manuscript can benefit from changes and clarification highlighted in my specific comments.
MAJOR:
1. The authors mentioned that “only variables with significant p-values in the univariate analysis were fitted and adjusted for in the final multivariate analysis.” While this is a good approach for academic statistical studies, the approach is seriously flawed in the current study aimed at producing research evidence for HIV programs and clinical practice. This approach of including only those variables in the model that are statistically significant is seriously flawed for a viral load suppression study.
a. It undermines the importance of factors scientifically known to influence Viral Suppression. The approach used by authors can lead to misspecification and overfitting of the model because the inclusion of individual variables only significantly associated with VS misses the interactions of variables and complex relationships, leading to bias and low generalizability of findings to other situations.
b. Omitting confounders of individual independent variables may highlight spurious relationships.
c. Further, not reporting negative findings results in loss of information. For instance, if the education of a PLWHIV is not a significant predictor of VS (yet included in the model) that is a valuable finding indicating the HIV clinical programs are effectively catering to all education levels, and that they should continue their prevention approaches with persons of all education levels. The authors must redo the analysis and use a theoretical approach to select the modifiable variables that are relevant to practice.
2. The choice of statistical methods is flawed and the authors have not justified it. The Chi-square analysis is redundant and potentially misleading because they do not account for confounders and may present spurious relationships. Chi-square is only justifiable in situations where assumptions of multivariable logistic regression are not met or there are too few observations to run multivariable analysis. The authors computed gross (unadjusted) as well as adjusted odds ratios using logistic regression. That eliminates the need for the chi-square analysis. The authors should justify the statistical methods’ selection and eliminate redundancy.
3. Write up of the results is incorrect for logistic regression model (detailed later).
Other specific comments:
TITLE:
4. In the title, there are a couple of flaws that must be addressed: (1) although technically OK, authors give away primary the implications/findings in the title “…Viral Suppression Supports Scaling-up Transition…” which may lower the incentive to read the full article and dissuade the reader from understanding the context. This may lead to oversimplification and may be misleading. (2) The title is too long. Consequently, it becomes difficult for the reader to find the central constructs of the research. The authors should consider revising the title to address these two points.
ABSTRACT:
5. Information critical for understanding the research is missing. The authors have not mentioned the statistical methods used. Also, the name of the data source and the year of data collection are missing.
6. The abstract should be revised to improve readability. Here are some examples.
· Line 46: VL should be spelled out at its first use.
· Line 46-49: The authors should rewrite the sentence “Regarding ART-regimen…” because they provide many numbers in parathesis impacting the readability.
· Line 49-50: In the sentence “Overall VS [95% CI] was 82.3% [81.5-83.2]… “ the [95% CI] seems out of place.
The INTRODUCTION:
7. The authors must spell out acronyms at their first use because many of them are less common. Here are some examples:
a. Line 99: The acronym “PLWHA” must be spelled out because it is the first occurrence in the manuscript.
b. Line 101-102: ODYSSEY and IMPAACT P1093…..DTG
8. In the closing paragraph, the purpose of the study should be elaborated, including that it aims to “identify factors independently associated with VS.” which is mentioned in the methods section when describing the logistic regression.
METHODS:
9. The methods section is missing a critical sub-section, a description of all of the variables in the study, and the operationalization/measure,emment of those variables.
10. In the study design, the authors must elaborate on which organization collected the data and whether it was primary data collection for this study or secondary use.
11. Line 143: The authors mention that RealTime PCR was used. For the benefit of the reader, the authors should provide a sentence about the difference between this PCR technique and conventional PCR, citing a published source.
12. Line 150: The software should be properly cited, providing the reference in the reference list.
RESULTS:
13. In Figure 1 and its interpretation, the authors use the term “non-virological suppression.” If the suppression is “non-virological” what is being suppressed? The authors need to use the correct term.
14. The results of Table 3 need to be re-written for clarity and technical soundness. For instance, when interpreting the adjusted ORs for age groups, the authors use the language only appropriate for interpreting linear regression when saying “In particular, for age, when compared to those 20-24 years, … were all negative associated with VS”; the odds of categorical variable cannot be stated as “associated”…They are low, high, or XYZ times.
DISCUSSION
The authors need to elaborate on their points in the paragraph on limitations. For instance, the authors mention that secondary data use may have introduced biases. This needs to be elaborated as to what type of biases. Otherwise, the statement is not very meaningless.
Comments on the Quality of English LanguageThe manuscript should be improved for readability and clarity.
Author Response
REVIEWER 2
- The authors mentioned that “only variables with significant p-values in the univariate analysis were fitted and adjusted for in the final multivariate analysis.” While this is a good approach for academic statistical studies, the approach is seriously flawed in the current study aimed at producing research evidence for HIV programs and clinical practice. This approach of including only those variables in the model that are statistically significant is seriously flawed for a viral load suppression study.
- It undermines the importance of factors scientifically known to influence Viral Suppression. The approach used by authors can lead to misspecification and overfitting of the model because the inclusion of individual variables only significantly associated with VS misses the interactions of variables and complex relationships, leading to bias and low generalizability of findings to other situations.
- Omitting confounders of individual independent variables may highlight spurious relationships.
- Further, not reporting negative findings results in loss of information. For instance, if the education of a PLWHIV is not a significant predictor of VS (yet included in the model) that is a valuable finding indicating the HIV clinical programs are effectively catering to all education levels, and that they should continue their prevention approaches with persons of all education levels. The authors must redo the analysis and use a theoretical approach to select the modifiable variables that are relevant to practice.
Response:
We thank the reviewer for these insightful comments. As the reviewer noted, these are programmatic data, and it is challenging in real-life to collect all the relevant variables such as education level that might be important in terms of viral suppression. In the case of this study, we successfully collected data only on 6 variables that were included in the regression model. For this reason, we did not perform a sophisticated analysis that include DAG (as previous done in one of our papers https://doi.org/10.1089/aid.2023.00). In fact, several of our previous papers on related topics used this simple but acceptable approach (http://dx.doi.org/10.1097/MD.0000000000033737, https://doi.org/10.1186/s12981-019-0252-0 ).
Moreover, as it can be observed in Table 3, all the 6 available variables (which are generally factors scientifically known to influence Viral Suppression in literature, including our previous studies) that were used in the univariate analysis were adjusted for in the multivariate model. To take into consideration the reviewer’s comment regarding the statistical approach used, we conducted a conditional backward stepwise regression model by fitting all the available variables (see Table 3). The p-value of the Hosmer and Lemeshow test for the goodness of fit is 0.907, which indicates a good regression model fit.
Even though the result from the model remains the same, in the present version of the manuscript, we have adopted the conditional backward stepwise regression model, for more clarity. The methods section was modified accordingly.
2. The choice of statistical methods is flawed and the authors have not justified it. The Chi-square analysis is redundant and potentially misleading because they do not account for confounders and may present spurious relationships. Chi-square is only justifiable in situations where assumptions of multivariable logistic regression are not met or there are too few observations to run multivariable analysis. The authors computed gross (unadjusted) as well as adjusted odds ratios using logistic regression. That eliminates the need for the chi-square analysis. The authors should justify the statistical methods’ selection and eliminate redundancy.
Response:
We thank the reviewer for this comment. Concerning the Chi-square analysis, we have deleted it from Table 1, as suggested. Table 2 presents the difference in proportion between those who suppressed their viral load and those who did not. We think that, as part of descriptive statistics, adding Chi-square might be useful, therefore, we plead with the reviewer to not delete it in Table 2. Moreover, we fully agree that Chi-square does not establish association, therefore our conclusions in terms of associated were solely based on data from the regression model.
3. Write up of the results is incorrect for logistic regression model (detailed later).
Response:
We have followed the recommendations of the reviewer and result presentation has been adjusted (see lines 318-341).
Other specific comments:
TITLE:
4. In the title, there are a couple of flaws that must be addressed: (1) although technically OK, authors give away primary the implications/findings in the title “…Viral Suppression Supports Scaling-up Transition…” which may lower the incentive to read the full article and dissuade the reader from understanding the context. This may lead to oversimplification and may be misleading. (2) The title is too long. Consequently, it becomes difficult for the reader to find the central constructs of the research. The authors should consider revising the title to address these two points.
Response:
We thank the reviewer for these important points. We have considered all these points, and the title is now: “Evaluation of Viral Suppression in Paediatric Populations: Implications for the Transition to Dolutegravir-Based Regimens in Cameroon: the CIPHER-ADOLA Study”
ABSTRACT:
5. Information critical for understanding the research is missing. The authors have not mentioned the statistical methods used. Also, the name of the data source and the year of data collection are missing.
Response:
Some of this information were not added because of the word count limitation. However, we have added the following sentences: “Data were collected from the database of nine reference laboratories in 2023. Conditional backward stepwise regression model was built to assess the predictors of VS.”
6. The abstract should be revised to improve readability. Here are some examples.
- Line 46: VL should be spelled out at its first use.
Response:
VL as well as other acronyms have been spelled out at their first use.
- Line 46-49: The authors should rewrite the sentence “RegardingART-regimen…” because they provide many numbers in parathesis impacting the readability.
Response:
This sentence has been rewritten and it now reads: “Regarding ART-regimen, 17% of children, 80% of adolescents, 83% of young-adults transitioned to dolutegravir (DTG)-based regimens”. We have removed the details in the bracket to improve readability.
- Line 49-50: In the sentence “Overall VS [95% CI] was 82.3% [81.5-83.2]… “ the [95% CI] seems out of place.
Response:
The 95% CI was deleted as suggested.
The INTRODUCTION:
7. The authors must spell out acronyms at their first use because many of them are less common. Here are some examples:
Line 99: The acronym “PLWHA” must be spelled out because it is the first occurrence in the manuscript.
Response:
PLWHA has been changes to PLHIV, which was already defined in a previous sentence.
Line 101-102: ODYSSEY and IMPAACT P1093…..DTG
Response:
These terms (as well as others) have been spelled out at their first use as recommended.
8. In the closing paragraph, the purpose of the study should be elaborated, including that it aims to “identify factors independently associated with VS.” which is mentioned in the methods section when describing the logistic regression.
Response:
The objective of the study has been adjusted and it now reads:
“…the objective of this study is to evaluate virological response and to identify factors independently associated with virological suppression among children, adolescents, and young adults…”
METHODS:
9. The methods section is missing a critical sub-section, a description of all of the variables in the study, and the operationalization/measure,emment of those variables.
Response:
In the new version, we have added a subsection “Description of study variables” in the methods section (see lines 150-168).
10. In the study design, the authors must elaborate on which organization collected the data and whether it was primary data collection for this study or secondary use.
Response
Some of this information were already in the “study population and data collection” subsection. Following the reviewer recommendations, we have elaborated more on this aspect in the present version (see lines 134-135 and 144-148).
11. Line 143: The authors mention that RealTime PCR was used. For the benefit of the reader, the authors should provide a sentence about the difference between this PCR technique and conventional PCR, citing a published source.
Response:
The 2 types of PCR platforms that were used for viral load testing were both conventional PCR. We have added the equipment names and link for clarity.
12. Line 150: The software should be properly cited, providing the reference in the reference list.
Response:
The information on the software has been updated, and the reference added.
RESULTS:
13. In Figure 1 and its interpretation, the authors use the term “non-virological suppression.” If the suppression is “non-virological” what is being suppressed? The authors need to use the correct term.
Response:
We thank the reviewer for this observation. The correct term is “Virological non-suppression”. We have corrected it in Fig.1 as well as in other parts of the manuscript.
14. The results of Table 3 need to be re-written for clarity and technical soundness. For instance, when interpreting the adjusted ORs for age groups, the authors use the language only appropriate for interpreting linear regression when saying “In particular, for age, when compared to those 20-24 years, … were all negative associated with VS”; the odds of categorical variable cannot be stated as “associated”…They are low, high, or XYZ times.
Response:
We thank the reviewer and we have rewritten this part following the reviewer’s recommendation in the section on regression model.
DISCUSSION
The authors need to elaborate on their points in the paragraph on limitations. For instance, the authors mention that secondary data use may have introduced biases. This needs to be elaborated as to what type of biases. Otherwise, the statement is not very meaningless.
Response:
We have elaborated a little more on the limitations, indicating the type of bias (see lines 452-455).
Reviewer 3 Report
Comments and Suggestions for Authors
Dear Authors,
Although the manuscript contains important information regarding the use of DTG-based regimens in children and adolescents, I have two major comments that should be considered.
Major comments:
First, please compare the results obtained with previously published analyses of viral suppression in people receiving different ART regimens, including large-scale clinical trials in different countries.
Second, please explicitly state the limitations of the study. It is important because it provides a recommendation for prioritising a rapid transition to a DTG-based regimen, which may be important for clinical practice. In general, I recommend rephrasing/deleting the following statement to clarify that "a rapid transition to DTG-based regimen" was shown in the current study and should be considered and confirmed in future large-scale clinical trials. Furthermore, not all DTG-based regimens considered in the study have high levels of viral suppression, for example ABC/3TC+DTG has an even lower rate of viral suppression than TDF/3TC-EFV/NVP with a lower number of participants. This clearly indicates that the statement "efforts towards eliminating paediatric AIDS should prioritise a rapid transition to a DTG-based regimen" is not supported by the results of the study.
English should be checked carefully, ideally with the help of a native speaker.
Comments on the Quality of English LanguageEnglish should be checked carefully, ideally with the help of a native speaker.
Author Response
REVIEWER 3
Although the manuscript contains important information regarding the use of DTG-based regimens in children and adolescents, I have two major comments that should be considered.
Response: We thank the reviewer for the time taken to review our manuscript.
Major comments:
First, please compare the results obtained with previously published analyses of viral suppression in people receiving different ART regimens, including large-scale clinical trials in different countries.
Response
Thank you for these comments. It should be noted that clinical trials in the paediatric populations is limited. However, in addition to the IMPAACT trial that was already cited in the manuscript, we have added more literatures (clinical trials and systematic review) in the discussion to allow comparisons (see lines 423-439).
Second, please explicitly state the limitations of the study. It is important because it provides a recommendation for prioritising a rapid transition to a DTG-based regimen, which may be important for clinical practice. In general, I recommend rephrasing/deleting the following statement to clarify that "a rapid transition to DTG-based regimen" was shown in the current study and should be considered and confirmed in future large-scale clinical trials. Furthermore, not all DTG-based regimens considered in the study have high levels of viral suppression, for example ABC/3TC+DTG has an even lower rate of viral suppression than TDF/3TC-EFV/NVP with a lower number of participants. This clearly indicates that the statement "efforts towards eliminating paediatric AIDS should prioritise a rapid transition to a DTG-based regimen" is not supported by the results of the study.
Response
We thank the reviewer for this important comment. We have given more clarifications on the limitations of the study by the end of the discussion section (see lines 452-455).
The following sentence was added: “This result should be considered and confirmed in future large-scale clinical trials and cohort studies”. See lines 437-438
The reviewer is correct about the observation ABC/3TC+DTG has an even lower rate of viral suppression than TDF/3TC-EFV/NVP. Following another reviewer’s suggestion, we have changed the variable “ART regimen” into “NRTI-backbone and anchor drug”. This was important because age is associated with VS and children cannot be treated with some regimen. Therefore, in this new version, from Table 3, it can be clearly seen that, TDF/3TC has a significantly higher odds compared to other backbones; and DTG also had a significantly higher odds compared to other anchor drugs.
It should be noted that TDF-backbone was used since several years in the Cameroonian context, and that the newer drug is DTG (for which limited data exist because of recent introduction), we emphasised the switch to DTG in our conclusions. However, we have softened the conclusions.
English should be checked carefully, ideally with the help of a native speaker.
Response: We have tried our best to improve the English in the present version of the manuscript.
Round 2
Reviewer 2 Report
Comments and Suggestions for Authors
I appreciate that the authors have appropriately edited the manuscript and addressed my comments.
Reviewer 3 Report
Comments and Suggestions for Authors
Dear Authors,
thank you very much for the manuscript and the changes that were made.
I recommend this paper to be accepted for the publication in Biomedicines Journal after careful check of English.
Comments on the Quality of English LanguageI recommend this paper to be accepted for the publication in Biomedicines Journal after careful check of English.